# Safe Adaptive Importance Sampling

**Sebastian U. Stich**
EPFL
sebastian.stich@epfl.ch

**Anant Raj**
Max Planck Institute for Intelligent Systems
anant.raj@tuebingen.mpg.de

**Martin Jaggi**
EPFL
martin.jaggi@epfl.ch

## Abstract

Importance sampling has become an indispensable strategy to speed up optimization algorithms for large-scale applications. Improved adaptive variants—using importance values defined by the complete gradient information which changes during optimization—enjoy favorable theoretical properties, but are typically computationally infeasible. In this paper we propose an efficient approximation of gradient-based sampling, which is based on safe bounds on the gradient. The proposed sampling distribution is (i) provably the *best sampling* with respect to the given bounds, (ii) always better than uniform sampling and fixed importance sampling and (iii) can efficiently be computed—in many applications at negligible extra cost. The proposed sampling scheme is generic and can easily be integrated into existing algorithms. In particular, we show that coordinate-descent (CD) and stochastic gradient descent (SGD) can enjoy significant a speed-up under the novel scheme. The proven efficiency of the proposed sampling is verified by extensive numerical testing.

## 1 Introduction

Modern machine learning applications operate on massive datasets. The algorithms that are used for data analysis face the difficult challenge to cope with the enormous amount of data or the vast dimensionality of the problems. A simple and well established strategy to reduce the computational costs is to split the data and to operate only on a small part of it, as for instance in coordinate descent (CD) methods and stochastic gradient (SGD) methods. These kind of methods are state of the art for a wide selection of machine learning, deep leaning and signal processing applications [9, 11, 35, 27]. The application of these schemes is not only motivated by their practical preformance, but also well justified by theory [18, 19, 2].

Deterministic strategies are seldom used for the data selection—examples are steepest coordinate descent [4, 34, 20] or screening algorithms [14, 15]. Instead, randomized selection has become ubiquitous, most prominently uniform sampling [27, 29, 7, 8, 28] but also non-uniform sampling based on a *fixed* distribution, commonly referred to as *importance sampling* [18, 19, 2, 33, 16, 6, 25, 24]. While these sampling strategies typically depend on the input data, they do not adapt to the information of the current parameters during optimization. In contrast, *adaptive* importance sampling strategies constantly re-evaluate the relative importance of each data point during training and thereby often surpass the performance of static algorithms [22, 5, 26, 10, 21, 23]. Common strategies are *gradient-based* sampling [22, 36, 37] (mostly for SGD) and *duality gap-based* sampling for CD [5, 23].

The drawbacks of adaptive strategies are twofold: often the provable theoretical guarantees can be worse than the complexity estimates for uniform sampling [23, 3] and often it is computationally

inadmissible to compute the optimal adaptive sampling distribution. For instance gradient based sampling requires the computation of the full gradient in each iteration [22, 36, 37]. Therefore one has to rely on approximations based on upper bounds [36, 37], or stale values [22, 1]. But in general these approximations can again be worse than uniform sampling.

This makes it necessary to develop adaptive strategies that can efficiently be computed in every iteration and that come with theoretical guarantees that show their advantage over fixed sampling.

**Our contributions.** In this paper we propose an efficient approximation of the gradient-based sampling in the sense that (i) it can efficiently be computed in every iteration, (ii) is provably better than uniform or fixed importance sampling and (iii) recovers the gradient-based sampling in the full-information setting. The scheme is completely generic and can easily be added as an improvement to both CD and SGD type methods.

As our key contributions, we

(1) show that gradient-based sampling in CD methods is theoretically better than the classical fixed sampling, the speed-up can reach a factor of the dimension $n$ (Section 2);
(2) propose a generic and efficient *adaptive importance sampling* strategy that can be applied in CD and SGD methods and enjoys favorable properties—such as mentioned above (Section 3);
(3) demonstrate how the novel scheme can efficiently be integrated in CD and SGD on an important class of structured optimization problems (Section 4);
(4) supply numerical evidence that the novel sampling performs well on real data (Section 5).

**Notation.** For $x \in \mathbb{R}^n$ define $[x]_i := \langle x, e_i \rangle$ with $e_i$ the standard unit vectors in $\mathbb{R}^n$. We abbreviate $\nabla_i f := [\nabla f]_i$. A convex function $f \colon \mathbb{R}^n \to \mathbb{R}$ with $L$-Lipschitz continuous gradient satisfies

$$f(x + \eta u) \leq f(x) + \eta \langle u, \nabla f(x) \rangle + \tfrac{\eta^2 L_u}{2} \|u\|^2 \qquad \forall x \in \mathbb{R}^n, \forall \eta \in \mathbb{R}, \qquad (1)$$

for every direction $u \in \mathbb{R}^n$ and $L_u = L$. A function with coordinate-wise $L_i$-Lipschitz continuous gradients[1] for constants $L_i > 0$, $i \in [n] := \{1, \dots, n\}$, satisfies (1) just along coordinate directions, i.e. $u = e_i$, $L_{e_i} = L_i$ for every $i \in [n]$. A function is coordinate-wise $L$-smooth if $L_i \leq L$ for $i = 1, \dots, n$. For convenience we introduce vector $l = (L_1, \dots, _n)^\top$ and matrix $\mathbf{L} = \mathrm{diag}(l)$. A probability vector $p \in \Delta^n := \{x \in \mathbb{R}^n_{\geq 0} \colon \|x\|_1 = 1\}$ defines a probability distribution $\mathcal{P}$ over $[n]$ and we denote by $i \sim p$ a sample drawn from $\mathcal{P}$.

## 2 Adaptive Importance Sampling with Full Information

In this section we argue that adaptive sampling strategies are theoretically well justified, as they can lead to significant improvements over static strategies. In our exhibition we focus first on CD methods, as we also propose a novel stepsize strategy for CD in this contribution. Then we revisit the results regarding stochastic gradient descent (SGD) already present in the literature.

### 2.1 Coordinate Descent with Adaptive Importance Sampling

We address general minimization problems $\min_x f(x)$. Let the objective $f \colon \mathbb{R}^n \to \mathbb{R}$ be convex with coordinate-wise $L_i$-Lipschitz continuous gradients. Coordinate descent methods generate sequences $\{x_k\}_{k \geq 0}$ of iterates that satisfy the relation

$$x_{k+1} = x_k - \gamma_k \nabla_{i_k} f(x_k) e_{i_k}. \qquad (2)$$

Here, the direction $i_k$ is either chosen deterministically (cyclic descent, steepest descent), or randomly picked according to a probability vector $p_k \in \Delta^n$. In the classical literature, the stepsize is often chosen such as to minimize the quadratic upper bound (1), i.e. $\gamma_k = L_{i_k}^{-1}$. In this work we propose to set $\gamma_k = \alpha_k [p_k]_{i_k}^{-1}$ where $\alpha_k$ does not depend on the chosen direction $i_k$. This leads to

directionally-unbiased updates, like it is common among SGD-type methods. It holds

$$\mathbb{E}_{i_k \sim \boldsymbol{p}_k} \left[ f(\boldsymbol{x}_{k+1}) \mid \boldsymbol{x}_k \right] \overset{(1)}{\leq} \mathbb{E}_{i_k \sim \boldsymbol{p}_k} \left[ f(\boldsymbol{x}_k) - \frac{\alpha_k}{[\boldsymbol{p}_k]_{i_k}} \left( \nabla_{i_k} f(\boldsymbol{x}_k) \right)^2 + \frac{L_{i_k} \alpha_k^2}{2[\boldsymbol{p}_k]_{i_k}^2} \left( \nabla_{i_k} f(\boldsymbol{x}_k) \right)^2 \mid \boldsymbol{x}_k \right]$$

$$= f(\boldsymbol{x}_k) - \alpha_k \|\nabla f(\boldsymbol{x}_k)\|_2^2 + \sum_{i=1}^n \frac{L_i \alpha_k^2}{2[\boldsymbol{p}_k]_i} \left( \nabla_i f(\boldsymbol{x}_k) \right)^2 . \tag{3}$$

In adaptive strategies we have the freedom to chose both variables $\alpha_k$ and $\boldsymbol{p}_k$ as we like. We therefore propose to chose them in such a way that they *minimize* the upper bound (3) in order to maximize the expected progress. The optimal $\boldsymbol{p}_k$ in (3) is independent of $\alpha_k$, but the optimal $\alpha_k$ depends on $\boldsymbol{p}_k$. We can state the following useful observation.

**Lemma 2.1.** *If $\alpha_k = \alpha_k(\boldsymbol{p}_k)$ is the minimizer of (3), then $\boldsymbol{x}_{k+1} := \boldsymbol{x}_k - \frac{\alpha_k}{[\boldsymbol{p}_k]_{i_k}} \nabla_{i_k} f(\boldsymbol{x}_k) \boldsymbol{e}_{i_k}$ satisfies*

$$\mathbb{E}_{i_k \sim \boldsymbol{p}_k} \left[ f(\boldsymbol{x}_{k+1}) \mid \boldsymbol{x}_k \right] \leq f(\boldsymbol{x}_k) - \frac{\alpha_k(\boldsymbol{p}_k)}{2} \|\nabla f(\boldsymbol{x}_k)\|_2^2 . \tag{4}$$

Consider two examples. In the first one we pick a sub-optimal, but very common [18] distribution:

**Example 2.2** ($L_i$-based sampling). *Let $\boldsymbol{p}_{\mathbf{L}} \in \Delta^n$ defined as $[\boldsymbol{p}_{\mathbf{L}}]_i = \frac{L_i}{\mathrm{Tr}[\mathbf{L}]}$ for $i \in [n]$, where $\mathbf{L} = \mathrm{diag}(L_1, \ldots, L_n)$. Then $\alpha_k(\boldsymbol{p}_{\mathbf{L}}) = \frac{1}{\mathrm{Tr}[\mathbf{L}]}$.*

The distribution $\boldsymbol{p}_{\mathbf{L}}$ is often referred to as (fixed) *importance* sampling. In the special case when $L_i = L$ for all $i \in [n]$, this boils down to uniform sampling.

**Example 2.3** (Optimal sampling[2]). *Equation (3) is minimized for probabilities $[\boldsymbol{p}_k^\star]_i = \frac{\sqrt{L_i}|\nabla_i f(\boldsymbol{x}_k)|}{\left\| \sqrt{\mathbf{L}} \nabla f(\boldsymbol{x}_k) \right\|_1}$ and $\alpha_k(\boldsymbol{p}_k^\star) = \frac{\|\nabla f(\boldsymbol{x}_k)\|_2^2}{\left\| \sqrt{\mathbf{L}} \nabla f(\boldsymbol{x}_k) \right\|_1^2}$. Observe $\frac{1}{\mathrm{Tr}[\mathbf{L}]} \leq \alpha_k(\boldsymbol{p}_k^\star) \leq \frac{1}{L_{\min}}$, where $L_{\min} := \min_{i \in [n]} L_i$.*

To prove this result, we rely on the following Lemma—the proof of which, as well as for the claims above, is deferred to Section A.1 of the appendix. Here $|\cdot|$ is applied entry-wise.

**Lemma 2.4.** *Define $V(\boldsymbol{p}, \boldsymbol{x}) := \sum_{i=1}^n \frac{L_i[\boldsymbol{x}]_i^2}{[\boldsymbol{p}]_i}$. Then $\arg\min_{\boldsymbol{p} \in \Delta^n} V(\boldsymbol{p}, \boldsymbol{x}) = \frac{|\sqrt{\mathbf{L}} \boldsymbol{x}|}{\left\| \sqrt{\mathbf{L}} \boldsymbol{x} \right\|_1}$.*

**The ideal adaptive algorithm.** We propose to chose the stepsize and the sampling distribution for CD as in Example 2.3. One iteration of the resulting CD method is illustrated in Algorithm 1. Our bounds on the expected one-step progress can be used to derive convergence rates of this algorithm with the standard techniques. This is exemplified in Appendix A.1. In the next Section 3 we develop a practical variant of the ideal algorithm.

**Efficiency gain.** By comparing the estimates provided in the examples above, we see that the expected progress of the proposed method is always at least as good as for the fixed sampling. For instance in the special case where $L = L_i$ for $i \in [n]$, the $L_i$-based sampling is just uniform sampling with $\alpha_k(\boldsymbol{p}_{\mathrm{unif}}) = \frac{1}{Ln}$. On the other hand $\alpha_k(\boldsymbol{p}_k^\star) = \frac{\|\nabla f(\boldsymbol{x}_k)\|_2^2}{L\|\nabla f(\boldsymbol{x}_k)\|_1^2}$, which can be $n$ times larger than $\alpha_k(\boldsymbol{p}_{\mathrm{unif}})$. The expected one-step progress in this extreme case coincides with the one-step progress of steepest coordinate descent [20].

## 2.2 SGD with Adaptive Sampling

SGD methods are applicable to objective functions which decompose as a sum

$$f(\boldsymbol{x}) = \frac{1}{n} \sum_{i=1}^n f_i(\boldsymbol{x}) \tag{5}$$

with each $f_i \colon \mathbb{R}^d \to \mathbb{R}$ convex. In previous work [22, 36, 37] is has been argued that the following gradient-based sampling $[\tilde{\boldsymbol{p}}_k^\star]_i = \frac{\|\nabla f_i(\boldsymbol{x}_k)\|_2}{\sum_{i=1}^n \|\nabla f_i(\boldsymbol{x}_k)\|_2}$ is optimal in the sense that it maximizes the expected progress (3). Zhao and Zhang [36] derive complexity estimates for composite functions. For non-composite functions it becomes easier to derive the complexity estimate. For completeness, we add this simpler proof in Appendix A.2.

| **Algorithm 1** Optimal sampling | **Algorithm 2** Proposed safe sampling | **Algorithm 3** Fixed sampling |
|---|---|---|
| *(compute full gradient)* | | |
| **Compute** $\nabla f(\boldsymbol{x}_k)$ | Update $\boldsymbol{\ell}, \boldsymbol{u}$ | |
| *(define optimal sampling)* | *(update l.- and u.-bounds)* | |
| Define $(\boldsymbol{p}_k^\star, \alpha_k^\star)$ as in Example 2.3 | *(compute safe sampling)* | *(define fixed sampling)* |
| $i_k \sim \boldsymbol{p}_k^\star$ | Define $(\hat{\boldsymbol{p}}_k, \hat{\alpha}_k)$ as in (7) | Define $(\boldsymbol{p}_L, \bar{\alpha})$ as in Example 2.2 |
| | $i_k \sim \hat{\boldsymbol{p}}_k$ | $i_k \sim \boldsymbol{p}_L$ |
| | **Compute** $\nabla_{i_k} f(\boldsymbol{x}_k)$ | **Compute** $\nabla_{i_k} f(\boldsymbol{x}_k)$ |
| $\boldsymbol{x}_{k+1} := \boldsymbol{x}_k - \frac{\alpha_k^\star}{[\boldsymbol{p}_k^\star]_{i_k}} \nabla_{i_k} f(\boldsymbol{x}_k)$ | $\boldsymbol{x}_{k+1} := \boldsymbol{x}_k - \frac{\hat{\alpha}_k}{[\hat{\boldsymbol{p}}_k]_{i_k}} \nabla_{i_k} f(\boldsymbol{x}_k)$ | $\boldsymbol{x}_{k+1} := \boldsymbol{x}_k - \frac{\bar{\alpha}}{[\boldsymbol{p}_L]_{i_k}} \nabla_{i_k} f(\boldsymbol{x}_k)$ |

Figure 1: CD with different sampling strategies. Whilst Alg. 1 requires to compute the full gradient, the compute operation in Alg. 2 is as cheap as for fixed importance sampling, Alg. 3. Defining the safe sampling $\hat{\boldsymbol{p}}_k$ requires $O(n \log n)$ time.

## 3 Safe Adaptive Importance Sampling with Limited Information

In the previous section we have seen that gradient-based sampling (Example 2.3) can yield a massive speed-up compared to a static sampling distribution (Example 2.2). However, sampling according to $\boldsymbol{p}_k^\star$ in CD requires the knowledge of the full gradient $\nabla f(\boldsymbol{x}_k)$ in each iteration. And likewise, sampling from $\tilde{\boldsymbol{p}}_k^\star$ in SGD requires the knowledge of the gradient norms of all components—both these operations are in general inadmissible, i.e. the compute cost would void all computational benefits of the iterative (stochastic) methods over full gradient methods.

However, it is often possible to efficiently compute *approximations* of $\boldsymbol{p}_k^\star$ or $\tilde{\boldsymbol{p}}_k^\star$ instead. In contrast to previous contributions, we here propose a *safe* way to compute such approximations. By this we mean that our approximate sampling is provably never worse than static sampling, and moreover, we show that our solution is the *best possible* with respect to the limited information at hand.

### 3.1 An Optimization Formulation for Sampling

Formally, we assume that we have in each iteration access to two vectors $\boldsymbol{\ell}_k, \boldsymbol{u}_k \in \mathbb{R}_{\geq 0}^n$ that provide safe upper and lower bounds on either the absolute values of the gradient entries ($[\boldsymbol{\ell}_k]_i \leq |\nabla_i f(\boldsymbol{x}_k)| \leq [\boldsymbol{u}_k]_i$) for CD, or of the gradient norms in SGD: ($[\boldsymbol{\ell}_k]_i \leq \|\nabla f_i(\boldsymbol{x}_k)\|_2 \leq [\boldsymbol{u}_k]_i$). We postpone the discussion of this assumption to Section 4, where we give concrete examples.

The minimization of the upper bound (3) amounts to the equivalent problem[3]

$$\min_{\alpha_k} \min_{\boldsymbol{p}_k \in \Delta^n} \left[ -\alpha_k \|\boldsymbol{c}_k\|_2^2 + \frac{\alpha_k^2}{2} V(\boldsymbol{p}_k, \boldsymbol{c}_k) \right] \quad \Leftrightarrow \quad \min_{\boldsymbol{p}_k \in \Delta^n} \frac{V(\boldsymbol{p}_k, \boldsymbol{c}_k)}{\|\boldsymbol{c}_k\|_2^2} \tag{6}$$

where $\boldsymbol{c}_k \in \mathbb{R}^n$ represents the *unknown* true gradient. That is, with respect to the bounds $\boldsymbol{\ell}_k, \boldsymbol{u}_k$, we can write $\boldsymbol{c}_k \in C_k := \{ \boldsymbol{x} \in \mathbb{R}^n \colon [\boldsymbol{\ell}_k]_i \leq [\boldsymbol{x}]_i \leq [\boldsymbol{u}_k]_i, i \in [n] \}$. In Example 2.3 we derived the optimal solution for a fixed $\boldsymbol{c}_k \in C_k$. However, this is not sufficient to find the optimal solution for an arbitrary $\boldsymbol{c}_k \in C_k$. Just computing the optimal solution for an arbitrary (but fixed) $\boldsymbol{c}_k \in C_k$ is unlikely to yield a good solution. For instance both extreme cases $\boldsymbol{c}_k = \boldsymbol{\ell}_k$ and $\boldsymbol{c}_k = \boldsymbol{u}_k$ (the latter choice is quite common, cf. [36, 23]) might be poor. This is demonstrated in the next example.

**Example 3.1.** *Let $\boldsymbol{\ell} = (1, 2)^\top$, $\boldsymbol{u} = (2, 3)^\top$, $\boldsymbol{c} = (2, 2)^\top$ and $L_1 = L_2 = 1$. Then $V\left(\frac{\boldsymbol{\ell}}{\|\boldsymbol{\ell}\|_1}, \boldsymbol{c}\right) = \frac{9}{4} \|\boldsymbol{c}\|_2^2$, $V\left(\frac{\boldsymbol{u}}{\|\boldsymbol{u}\|_1}, \boldsymbol{c}\right) = \frac{25}{12} \|\boldsymbol{c}\|_2^2$, whereas for uniform sampling $V\left(\frac{\boldsymbol{c}}{\|\boldsymbol{c}\|_1}, \boldsymbol{c}\right) = 2 \|\boldsymbol{c}\|_2^2$.*

**The proposed sampling.** As a consequence of these observations, we propose to solve the following optimization problem to find the best sampling distribution with respect to $C_k$:

$$v_k := \min_{\boldsymbol{p} \in \Delta^n} \max_{\boldsymbol{c} \in C_k} \frac{V(\boldsymbol{p}, \boldsymbol{c})}{\|\boldsymbol{c}\|_2^2}, \qquad \text{and to set} \qquad (\alpha_k, \boldsymbol{p}_k) := \left(\frac{1}{v_k}, \hat{\boldsymbol{p}}_k\right), \tag{7}$$

where $\hat{\boldsymbol{p}}_k$ denotes a solution of (7). The resulting algorithm for CD is summarized in Alg. 2.

In the remainder of this section we discuss the properties of the solution $\hat{\boldsymbol{p}}_k$ (Theorem 3.2) and how such a solution can be efficiently be computed (Theorem 3.4, Algorithm 4).

## 3.2 Proposed Sampling and its Properties

**Theorem 3.2.** *Let $(\hat{\boldsymbol{p}}, \hat{\boldsymbol{c}}) \in \Delta^n \times \mathbb{R}_{\geq 0}^n$ denote a solution of* (7). *Then $L_{\min} \leq v_k \leq \mathrm{Tr}\,[\mathbf{L}]$ and*

*(i)* $\displaystyle \max_{\boldsymbol{c} \in C_k} \frac{V(\hat{\boldsymbol{p}}, \boldsymbol{c})}{\|\boldsymbol{c}\|_2^2} \leq \max_{\boldsymbol{c} \in C_k} \frac{V(\boldsymbol{p}, \boldsymbol{c})}{\|\boldsymbol{c}\|_2^2}, \forall \boldsymbol{p} \in \Delta^n;$          *($\hat{\boldsymbol{p}}$ has the best worst-case guarantee)*

*(ii)* $V(\hat{\boldsymbol{p}}, \boldsymbol{c}) \leq \mathrm{Tr}\,[\mathbf{L}] \cdot \|\boldsymbol{c}\|_2^2, \forall \boldsymbol{c} \in C_k.$       *($\hat{\boldsymbol{p}}$ is always better than $L_i$-based sampling)*

**Remark 3.3.** *In the special case $L_i = L$ for all $i \in [n]$, the $L_i$-based sampling boils down to uniform sampling (Example 2.2) and $\hat{\boldsymbol{p}}$ is better than uniform sampling: $V(\hat{\boldsymbol{p}}, \boldsymbol{c}) \leq Ln \|\boldsymbol{c}\|_2^2, \forall \boldsymbol{c} \in C_k$.*

*Proof.* Property *(i)* is an immediate consequence of (7). Moreover, observe that the $L_i$-based sampling $\boldsymbol{p}_L$ is a feasible solution in (7) with value $\frac{V(\boldsymbol{p}_L, \boldsymbol{c})}{\|\boldsymbol{c}\|_2^2} \equiv \mathrm{Tr}\,[\mathbf{L}]$ for all $\boldsymbol{c} \in C_k$. Hence

$$L_{\min} \leq \frac{\|\sqrt{\mathbf{L}}\boldsymbol{c}\|_1^2}{\|\boldsymbol{c}\|_2^2} \overset{2.4}{=} \min_{\boldsymbol{p} \in \Delta^n} \frac{V(\boldsymbol{p}, \boldsymbol{c})}{\|\boldsymbol{c}\|_2^2} \leq \frac{V(\hat{\boldsymbol{p}}, \boldsymbol{c})}{\|\boldsymbol{c}\|_2^2} \overset{(*)}{\leq} \frac{V(\hat{\boldsymbol{p}}, \hat{\boldsymbol{c}})}{\|\hat{\boldsymbol{c}}\|_2^2} \overset{(7)}{\leq} \max_{\boldsymbol{c} \in C_k} \frac{V(\boldsymbol{p}_L, \boldsymbol{c})}{\|\boldsymbol{c}\|_2^2} = \mathrm{Tr}\,[\mathbf{L}] \ , \quad (8)$$

for all $\boldsymbol{c} \in C_k$, thus $v_k \in [L_{\min}, \mathrm{Tr}\,[\mathbf{L}]]$ and *(ii)* follows. We prove inequality $(*)$ in the appendix, by showing that min and max can be interchanged in (7). $\qquad\square$

**A geometric interpretation.** We show in Appendix B that the optimization problem (7) can equivalently be written as $\sqrt{v_k} = \max_{\boldsymbol{c} \in C_k} \frac{\|\sqrt{\mathbf{L}}\boldsymbol{c}\|_1}{\|\boldsymbol{c}\|_2} = \max_{\boldsymbol{c} \in C_k} \frac{\langle \sqrt{\boldsymbol{l}}, \boldsymbol{c} \rangle}{\|\boldsymbol{c}\|_2}$, where $[\boldsymbol{l}]_i = L_i$ for $i \in [n]$. The maximum is thus attained for vectors $\boldsymbol{c} \in C_k$ that minimize the angle with the vector $\boldsymbol{l}$.

**Theorem 3.4.** *Let $\boldsymbol{c} \in C_k$, $\boldsymbol{p} = \frac{\sqrt{\mathbf{L}}\boldsymbol{c}}{\|\sqrt{\mathbf{L}}\boldsymbol{c}\|_1}$ and denote $m = \|\boldsymbol{c}\|_2^2 \cdot \|\sqrt{\mathbf{L}}\boldsymbol{c}\|_1^{-1}$. If*

$$[\boldsymbol{c}]_i = \begin{cases} [\boldsymbol{u}_k]_i & \text{if } [\boldsymbol{u}_k]_i \leq \sqrt{L_i}m \,, \\ [\boldsymbol{\ell}_k]_i & \text{if } [\boldsymbol{\ell}_k]_i \geq \sqrt{L_i}m \,, \\ \sqrt{L_i}m & \text{otherwise,} \end{cases} \qquad \forall i \in [n] \,, \qquad (9)$$

*then $(\boldsymbol{p}, \boldsymbol{c})$ is a solution to* (7). *Moreover, such a solution can be computed in time $O(n \log n)$.*

*Proof.* This can be proven by examining the optimality conditions of problem (7). This is deferred to Section B.1 of the appendix. A procedure that computes such a solution is depicted in Algorithm 4. The algorithm makes extensive use of (9). For simplicity, assume first $\mathbf{L} = \mathbf{I}_n$ for now. In each iteration $t$, a potential solution vector $\boldsymbol{c}_t$ is proposed, and it is verified whether this vector satisfies all optimality conditions. In Algorithm 4, $\boldsymbol{c}_t$ is just implicit, with $[\boldsymbol{c}_t]_i = [\boldsymbol{c}]_i$ for decided indices $i \in D$ and $[\boldsymbol{c}_t]_i = [\sqrt{\mathbf{L}}m]_i$ for undecided indices $i \notin D$. After at most $n$ iterations a valid solution is found. By sorting the components of $\sqrt{\mathbf{L}^{-1}}\boldsymbol{\ell}_k$ and $\sqrt{\mathbf{L}^{-1}}\boldsymbol{u}_k$ by their magnitude, at most a linear number of inequality checks in (9) have to be performed in total. Hence the running time is dominated by the $O(n \log n)$ complexity of the sorting algorithm. A formal proof is given in the appendix. $\qquad\square$

---

**Algorithm 4** Computing the Safe Sampling for Gradient Information $\boldsymbol{\ell}, \boldsymbol{u}$

1: **Input:** $\mathbf{0}_n \leq \boldsymbol{\ell} \leq \boldsymbol{u}$, $\mathbf{L}$, **Initialize:** $\boldsymbol{c} = \mathbf{0}_n$, $u = 1$, $\ell = n$, $D = \emptyset$.
2: $\boldsymbol{\ell}^{\mathrm{sort}} := \mathrm{sort\_asc}(\sqrt{\mathbf{L}^{-1}}\boldsymbol{\ell})$, $\boldsymbol{u}^{\mathrm{sort}} := \mathrm{sort\_asc}(\sqrt{\mathbf{L}^{-1}}\boldsymbol{u})$, $m = \max(\boldsymbol{\ell}^{\mathrm{sort}})$
3: **while** $u \leq \ell$ **do**
4:     **if** $[\boldsymbol{\ell}^{\mathrm{sort}}]_\ell > m$ **then**         *(largest undecided lower bound is violated)*
5:         Set corresponding $[\boldsymbol{c}]_{\mathrm{index}} := [\sqrt{\mathbf{L}}\boldsymbol{\ell}^{\mathrm{sort}}]_\ell$; $\ell := \ell - 1$; $D := D \cup \{\mathrm{index}\}$
6:     **else if** $[\boldsymbol{u}^{\mathrm{sort}}]_u < m$ **then**      *(smallest undecided upper bound is violated)*
7:         Set corresponding $[\boldsymbol{c}]_{\mathrm{index}} := [\sqrt{\mathbf{L}}\boldsymbol{u}^{\mathrm{sort}}]_u$; $u := u + 1$; $D := D \cup \{\mathrm{index}\}$
8:     **else**
9:         **break**                     *(no constraints are violated)*
10:    **end if**
11:    $m := \|\boldsymbol{c}\|_2^2 \cdot \|\sqrt{\mathbf{L}}\boldsymbol{c}\|_1^{-1}$            *(update $m$ as in (9))*
12: **end while**
13: Set $[\boldsymbol{c}]_i := \sqrt{L_i}m$ for all $i \notin D$ and **Return** $\left( \boldsymbol{c}, \boldsymbol{p} = \frac{\sqrt{\mathbf{L}}\boldsymbol{c}}{\|\sqrt{\mathbf{L}}\boldsymbol{c}\|_1}, v = \frac{\|\sqrt{\mathbf{L}}\boldsymbol{c}\|_1^2}{\|\boldsymbol{c}\|_2^2} \right)$

---

**Competitive Ratio.** We now compare the proposed sampling distribution $\hat{p}_k$ with the optimal sampling solution in *hindsight*. We know that if the true (gradient) vector $\tilde{c} \in C_k$ would be given to us, then the corresponding optimal probability distribution would be $p^\star(\tilde{c}) = \frac{\sqrt{\mathbf{L}}\tilde{c}}{\|\sqrt{\mathbf{L}}\tilde{c}\|_1}$ (Example 2.3). Thus, for this $\tilde{c}$ we can now analyze the ratio $\frac{V(\hat{p}_k, \tilde{c})}{V(p^\star(\tilde{c}), \tilde{c})}$. As we are interested in the worst case ratio among all possible candidates $\tilde{c} \in C_k$, we define

$$\rho_k := \max_{c \in C_k} \frac{V(\hat{p}, c)}{V(p^\star(c), c)} = \max_{c \in C_k} \frac{V(\hat{p}, c)}{\|\sqrt{\mathbf{L}}c\|_1^2}. \tag{10}$$

**Lemma 3.5.** *Let* $w_k := \min_{c \in C_k} \frac{\|\sqrt{\mathbf{L}}c\|_1^2}{\|c\|_2^2}$. *Then* $L_{\min} \leq w_k \leq v_k$, *and* $\rho_k \leq \frac{v_k}{w_k} (\leq \frac{v_k}{L_{\min}})$.

**Lemma 3.6.** *Let* $\gamma \geq 1$. *If* $[C_k]_i \cap \gamma[C_k]_i = \emptyset$ *and* $\gamma^{-1}[C_k]_i \cap [C_k]_i = \emptyset$ *for all* $i \in [n]$ *(here* $[C_k]_i$ *denotes the projection on the $i$-th coordinate), then* $\rho_k \leq \gamma^4$.

These two lemma provide bounds on the competitive ratio. Whilst Lemma 3.6 relies on a relative accuracy condition, Lemma 3.5 can always be applied. However, the corresponding minimization problem is non-convex. Note that knowledge of $\rho_k$ is not needed to run the algorithm.

# 4 Example Safe Gradient Bounds

In this section, we argue that for a large class of objective functions of interest in machine learning, suitable safe upper and lower bounds $\ell, u$ on the gradient along every coordinate direction can be estimated and maintained efficiently during optimization. A similar argument can be given for the efficient approximation of component wise gradient norms in finite sum objective based stochastic gradient optimization.

As the guiding example, we will here showcase the training of generalized linear models (GLMs) as e.g. in regression, classification and feature selection. These models are formulated in terms of a given data matrix $\mathbf{A} \in \mathbb{R}^{d \times n}$ with columns $a_i \in \mathbb{R}^d$ for $i \in [n]$.

**Coordinate Descent - GLMs with Arbitrary Regularizers.** Consider general objectives of the form $f(x) := h(\mathbf{A}x) + \sum_{i=1}^n \psi_i([x]_i)$ with an arbitrary convex separable regularizer term given by the $\psi_i \colon \mathbb{R} \to \mathbb{R}$ for $i \in [n]$. A key example is when $h \colon \mathbb{R}^d \to \mathbb{R}$ describes the *least-squares* regression objective $h(\mathbf{A}x) = \frac{1}{2} \|\mathbf{A}x - b\|_2^2$ for a $b \in \mathbb{R}^d$. Using that this $h$ is twice differentiable with $\nabla^2 h(\mathbf{A}x) = \mathbf{I}_n$, it is easy to see that we can track the evolution of all gradient entries, when performing CD steps, as follows:

$$\nabla_i f(x_{k+1}) - \nabla_i f(x_k) = \gamma_k \langle a_i, a_{i_k} \rangle, \quad \forall i \neq i_k. \tag{11}$$

for $i_k$ being the coordinate changed in step $k$ (here we also used the separability of the regularizer).

Therefore, all gradient changes can be tracked exactly if the inner products of all datapoints are available, or approximately if those inner products can be upper and lower bounded. For computational efficiency, we in our experiments simply use Cauchy-Schwarz $|\langle a_i, a_{i_k} \rangle| \leq \|a_i\| \cdot \|a_{i_k}\|$. This results in safe upper and lower bounds $[\ell_{k+1}]_i \leq \nabla_i f(x_{k+1}) \leq [u_{k+1}]_i$ for all inactive coordinates $i \neq i_k$. (For the active coordinate $i_k$ itself one observes the true value without uncertainty). These bounds can be updated in linear time $O(n)$ in every iteration.

For general smooth $h$ (again with arbitrary separable regularizers $\psi_i$), (11) can readily be extended to hold [32, Lemma 4.1], the inner product change term becoming $\langle a_i, \nabla^2 f(\mathbf{A}\tilde{x}) a_{i_k} \rangle$ instead, when assuming $h$ is twice-differentiable. Here $\tilde{x}$ will be an element of the line segment $[x_k, x_{k+1}]$.

**Stochastic Gradient Descent - GLMs.** We now present a similar result for finite sum problems (5) for the use in SGD based optimization, that is $f(x) := \frac{1}{n} \sum_{i=1}^n f_i(x) = \frac{1}{n} \sum_{i=1}^n h_i(a_i^\top x)$.

**Lemma 4.1.** *Consider* $f \colon \mathbb{R}^d \to \mathbb{R}$ *as above, with twice differentiable* $h_i \colon \mathbb{R} \to \mathbb{R}$. *Let* $x_k, x_{k+1} \in \mathbb{R}^d$ *denote two successive iterates of SGD, i.e.* $x_{k+1} := x_k - \eta_k a_{i_k} \nabla h_{i_k}(a_{i_k}^\top x_k) = x_k + \gamma_k a_{i_k}$. *Then there exists* $\tilde{x} \in \mathbb{R}^d$ *on the line segment between* $x_k$ *and* $x_{k+1}$, $\tilde{x} \in [x_k, x_{k+1}]$ *with*

$$\nabla f_i(x_{k+1}) - \nabla f_i(x_k) = \gamma_k \nabla^2 h_i(a_i^\top \tilde{x}) \langle a_i, a_{i_k} \rangle a_i, \quad \forall i \neq i_k. \tag{12}$$

This leads to safe upper and lower bounds for the norms of the partial gradient, $[\boldsymbol{\ell}_k]_i \leq \|\nabla f_i(\boldsymbol{x}_k)\|_2 \leq [\boldsymbol{u}_k]_i$, that can be updated in linear time $O(n)$, analogous to the coordinate case discussed above.[4]

We note that there are many other ways to track safe gradient bounds for relevant machine learning problems, including possibly more tight ones. We here only illustrate the simplest variants, highlighting the fact that our new sampling procedure works for any safe bounds $\boldsymbol{\ell}, \boldsymbol{u}$.

**Computational Complexity.** In this section, we have demonstrated how safe upper and lower bounds $\boldsymbol{\ell}, \boldsymbol{u}$ on the gradient information can be obtained for GLMs, and argued that these bounds can be updated in time $O(n)$ per iteration of CD and SGD. The computation of the proposed sampling takes $O(n \log n)$ time (Theorem 3.4). Hence, the introduced overhead in Algorithm 2 compared to fixed sampling (Algorithm 3) is of the order $O(n \log n)$ in every iteration. The computation of one coordinate of the gradient, $\nabla_{i_k} f(\boldsymbol{x}_k)$, takes time $\Theta(d)$ for general data matrices. Hence, when $d = \Omega(n)$, the introduced overhead reduces to $O(\log n)$ per iteration.

# 5   Empirical Evaluation

In this section we evaluate the empirical performance of our proposed adaptive sampling scheme on relevant machine learning tasks. In particular, we illustrate performance on generalized linear models with $L1$ and $L2$ regularization, as of the form (5),

$$\min_{\boldsymbol{x} \in \mathbb{R}^d} \frac{1}{n} \sum_{i=1}^{n} h_i(\boldsymbol{a}_i^\top \boldsymbol{x}) + \lambda \cdot r(\boldsymbol{x}) \tag{13}$$

We use square loss, squared hinge loss as well as logistic loss for the data fitting terms $h_i$, and $\|\boldsymbol{x}\|_1$ and $\|\boldsymbol{x}\|_2^2$ for the regularizer $r(\boldsymbol{x})$. The datasets used in the evaluation are *rcv1*, *real-sim* and *news20*.[5] The *rcv1* dataset consists of $20{,}242$ samples with $47{,}236$ features, *real-sim* contains $72{,}309$ datapoints and $20{,}958$ features and *news20* contains $19{,}996$ datapoints and $1{,}355{,}191$ features. For all datasets we set unnormalized features with all the non-zero entries set to 1 (bag-of-words features). By *real-sim'* and *rcv1'* we denote a subset of the data chosen by randomly selecting 10,000 features and 10,000 datapoints. By *news20'* we denote a subset of the data chose by randomly selecting 15% of the features and 15% of the datapoints. A regularization parameter $\lambda = 0.1$ is used for all experiments.

Our results show the evolution of the optimization objective over time or number of epochs (an epoch corresponding to $n$ individual updates). To compute safe lower and upper bounds we use the methods presented in Section 4 with no special initialization, i.e. $\boldsymbol{\ell}_0 = \boldsymbol{0}_n$, $\boldsymbol{u}_0 = \boldsymbol{\infty}_n$.

**Coordinate Descent.** In Figure 2 we compare the effect of the fixed stepsize $\alpha_k = \frac{1}{Ln}$ (denoted as "small") vs. the time varying optimal stepsize (denoted as "big") as discussed in Section 2. Results are shown for optimal sampling $\boldsymbol{p}_k^\star$ (with optimal stepsize $\alpha_k(\boldsymbol{p}_k^\star)$, cf. Example 2.3), our proposed sampling $\hat{\boldsymbol{p}}_k$ (with optimal stepsize $\alpha_k(\hat{\boldsymbol{p}}_k) = v_k^{-1}$, cf. (7)) and uniform sampling (with optimal stepsize $\alpha_k(\boldsymbol{p}_{\mathbf{L}}) = \frac{1}{Ln}$, as here $\mathbf{L} = L\mathbf{I}_n$, cf. Example 2.2). As the experiment aligns with theory—confirming the advantage of the varying "big" stepsizes—we only show the results for Algorithms 1–3 in the remaining plots.

Performance for squared hinge loss, as well as logistic regression with $L1$ and $L2$ regularization is presented in Figure 3 and Figure 4 respectively. In Figures 5 and 6 we report the iteration complexity vs. accuracy as well as timing vs. accuracy results on the full dataset for coordinate descent with square loss and $L1$ (Lasso) and $L2$ regularization (Ridge).

**Theoretical Sampling Quality.** As part of the CD performance results in Figures 2–6 we include an additional evolution plot on the bottom of each figure to illustrate the values $v_k$ which determine the stepsize ($\hat{\alpha}_k = v_k^{-1}$) for the proposed Algorithm 2 (blue) and the optimal stepsizes of Algorithm 1 (black) which rely on the full gradient information. The plots show the normalized values $\frac{v_k}{\text{Tr}[\mathbf{L}]}$, i.e. the relative improvement over $L_i$-based importance sampling. The results show that despite only relying on very loose safe gradient bounds, the proposed adaptive sampling is able to strongly benefit from the additional information.

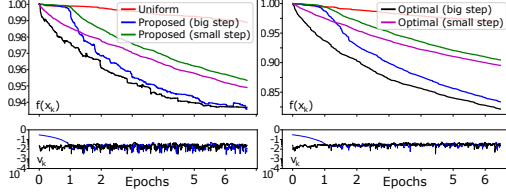

(a) *rcv1'*, $L1$ reg.  (b) *rcv1'*, $L2$ reg.

Figure 2: (CD, square loss) Fixed vs. adaptive sampling strategies, and dependence on stepsizes. With "big" $\alpha_k = v_k^{-1}$ and "small" $\alpha_k = \frac{1}{\text{Tr}[\mathbf{L}]}$.

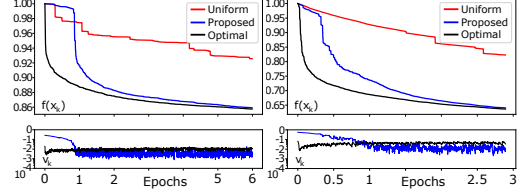

(a) *rcv1'*, $L1$ reg.  (b) *real-sim'*, $L2$ reg.

Figure 3: (CD, squared hinge loss) Function value vs. number of iterations for optimal stepsize $\alpha_k = v_k^{-1}$.

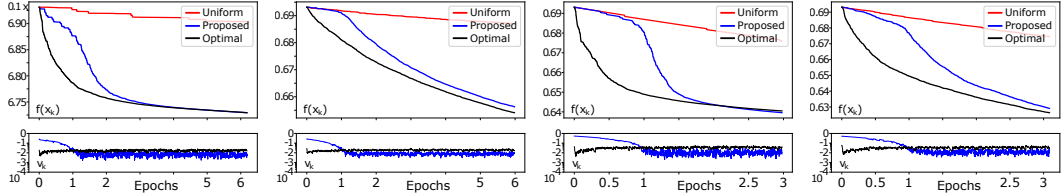

(a) *rcv1'*, $L1$ reg.  (b) *rcv1'*, $L2$ reg.  (c) *real-sim'*, $L1$ reg.  (d) *real-sim'*, $L2$ reg.

Figure 4: (CD, logistic loss) Function value vs. number of iterations for different sampling strategies. Bottom: Evolution of the value $v_k$ which determines the optimal stepsize ($\hat{\alpha}_k = v_k^{-1}$). The plots show the normalized values $\frac{v_k}{\text{Tr}[\mathbf{L}]}$, i.e. the relative improvement over $L_i$-based importance sampling.

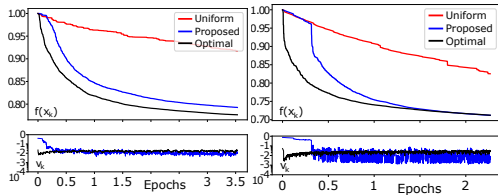

(a) *rcv1*, $L1$ reg.  (b) *real-sim*, $L1$ reg.

Figure 5: (CD, square loss) Function value vs. number of iterations on the full datasets.

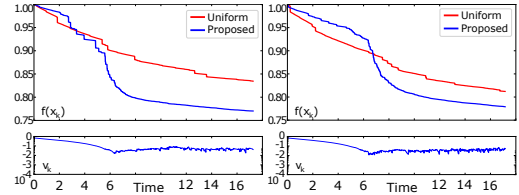

(a) *real-sim*, $L1$ reg.  (b) *real-sim*, $L2$ reg.

Figure 6: (CD, square loss) Function value vs. clock time on the full datasets. (Data for the optimal sampling omitted, as this strategy is not competitive time-wise.)

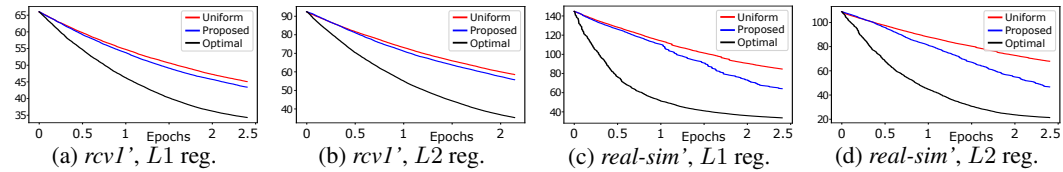

(a) *rcv1'*, $L1$ reg.  (b) *rcv1'*, $L2$ reg.  (c) *real-sim'*, $L1$ reg.  (d) *real-sim'*, $L2$ reg.

Figure 7: (SGD, square loss) Function value vs. number of iterations.

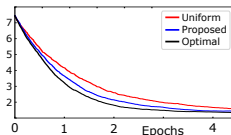

(a) *news20'*, $L1$ reg.

Figure 8: (SGD, square loss) Function value vs. number of iterations.

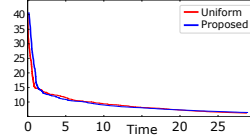

(a) *news20'*, $L1$ reg.

Figure 9: (SGD square loss) Function value vs. clock time.

**Stochastic Gradient Descent.** Finally, we also evaluate the performance of our approach when used within SGD with $L1$ and $L2$ regularization and square loss. In Figures 7–8 we report the iteration complexity vs. accuracy results and in Figure 9 the timing vs. accuracy results. The time units in Figures 6 and 9 are not directly comparable, as the experiments were conducted on different machines.

We observe that on all three datasets SGD with the optimal sampling performs only slightly better than uniform sampling. This is in contrast with the observations for CD, where the optimal sampling yields a significant improvement. Consequently, the effect of the proposed sampling is less pronounced in the three SGD experiments.

**Summary.** The main findings of our experimental study can be summarized as follows:

- **Adaptive importance sampling significantly outperforms fixed importance sampling in iterations and time.** The results show that (i) convergence in terms of iterations is almost as good as for the optimal (but not efficiently computable) gradient-based sampling and (ii) the introduced computational overhead is small enough to outperform fixed importance sampling in terms of total computation time.

- **Adaptive sampling requires adaptive stepsizes.** The adaptive stepsize strategies of Algorithms 1 and 2 allow for much faster convergence than conservative fixed-stepsize strategies. In the experiments, the measured value $v_k$ was always significantly below the worst case estimate, in alignment with the observed convergence.

- **Very loose safe gradient bounds are sufficient.** Even the bounds derived from the the very naïve gradient information obtained by estimating scalar products resulted in significantly better sampling than using no gradient information at all. Further, no initialization of the gradient estimates is needed (at the beginning of the optimization process the proposed adaptive method performs close to the fixed sampling but accelerates after just one epoch).

# 6 Conclusion

In this paper we propose a safe adaptive importance sampling scheme for CD and SGD algorithms. We argue that optimal gradient-based sampling is theoretically well justified. To make the computation of the adaptive sampling distribution computationally tractable, we rely on safe lower and upper bounds on the gradient. However, in contrast to previous approaches, we use these bounds in a novel way: in each iteration, we formulate the problem of picking the optimal sampling distribution as a convex optimization problem and present an efficient algorithm to compute the solution. The novel sampling provably performs better than any fixed importance sampling—a guarantee which could not be established for previous samplings that were also derived from safe lower and upper bounds.

The computational cost of the proposed scheme is of the order $O(n \log n)$ per iteration—this is on many problems comparable with the cost to evaluate a single component (coordinate, sum-structure) of the gradient, and the scheme can thus be implemented at no extra computational cost. This is verified by timing experiments on real datasets.

We discussed one simple method to track the gradient information in GLMs during optimization. However, we feel that the machine learning community could profit from further research in that direction, for instance by investigating how such safe bounds can efficiently be maintained on more complex models. Our approach can immediately be applied when the tracking of the gradient is delegated to other machines in a distributed setting, like for instance in [1].

## Footnotes

[1] $|\nabla_i f(x + \eta e_i) - \nabla_i f(x)| \leq L_i |\eta|, \quad \forall x \in \mathbb{R}^n, \forall \eta \in \mathbb{R}.$

[2] Here "optimal" refers to the fact that $\boldsymbol{p}_k^\star$ is optimal with respect to the given model (1) of the objective function. If the model is not accurate, there might exist a sampling that yields larger expected progress on $f$.

[3] Although only shown here for CD, an equivalent optimization problem arises for SGD methods, cf. [36].

[4]Here we use the efficient representation $\nabla f_i(\boldsymbol{x}) = \theta(\boldsymbol{x}) \cdot \boldsymbol{a}_i$ for $\theta(\boldsymbol{x}) \in \mathbb{R}$.

[5]All data are available at www.csie.ntu.edu.tw/~cjlin/libsvmtools/datasets/

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
