[Supplementary Material]

# Safe Adaptive Importance Sampling

**Sebastian U. Stich**
EPFL
sebastian.stich@epfl.ch

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

**Notation.** For $\boldsymbol{x} \in \mathbb{R}^n$ define $[\boldsymbol{x}]_i := \langle \boldsymbol{x}, \boldsymbol{e}_i \rangle$ with $\boldsymbol{e}_i$ the standard unit vectors in $\mathbb{R}^n$. We abbreviate $\nabla_i f := [\nabla f]_i$. A convex function $f \colon \mathbb{R}^n \to \mathbb{R}$ with $L$-Lipschitz continuous gradient satisfies

$$f(\boldsymbol{x} + \eta \boldsymbol{u}) \leq f(\boldsymbol{x}) + \eta \langle \boldsymbol{u}, \nabla f(\boldsymbol{x}) \rangle + \tfrac{\eta^2 L_{\boldsymbol{u}}}{2} \|\boldsymbol{u}\|^2 \qquad \forall \boldsymbol{x} \in \mathbb{R}^n, \forall \eta \in \mathbb{R}, \qquad (1)$$

for every direction $\boldsymbol{u} \in \mathbb{R}^n$ and $L_{\boldsymbol{u}} = L$. A function with coordinate-wise $L_i$-Lipschitz continuous gradients[1] for constants $L_i > 0$, $i \in [n] := \{1, \ldots, n\}$, satisfies (1) just along coordinate directions, i.e. $\boldsymbol{u} = \boldsymbol{e}_i$, $L_{\boldsymbol{e}_i} = L_i$ for every $i \in [n]$. A function is coordinate-wise $L$-smooth if $L_i \leq L$ for $i = 1, \ldots, n$. For convenience we introduce vector $\boldsymbol{l} = (L_1, \ldots, n)^\top$ and matrix $\mathbf{L} = \mathrm{diag}(\boldsymbol{l})$. A probability vector $\boldsymbol{p} \in \Delta^n := \{\boldsymbol{x} \in \mathbb{R}^n_{\geq 0} \colon \|\boldsymbol{

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

(\boldsymbol{x} + \eta \boldsymbol{e}_i) - \nabla_i f(\boldsymbol{x})| \leq L_i |\eta|, \quad \forall \boldsymbol{x} \in \mathbb{R}^n, \forall \eta \in \mathbb{R}.$

[2]Here "optimal" refers to the fact that $\boldsymbol{p}_k^\star$ is optimal with respect to the given model (1) of the objective function. If the model is not accurate, there might exist a sampling that yields larger expected progress on $f$.

[3]Although only shown here for CD, an equivalent optimization problem arises for SGD methods, cf. [36].

[4]Here we use the efficient representation $\nabla f_i(\boldsymbol{x}) = \theta(\boldsymbol{x}) \cdot \boldsymbol{a}_i$ for $\theta(\boldsymbol{x}) \in \mathbb{R}$.

[5]All data are available at www.csie.ntu.edu.tw/~cjlin/libsvmtools/datasets/

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

# Appendix

## A  Efficiency of Adaptive Importance Sampling

In this section of the appendix we present the missing proofs from the main text and also add some additional comments.

### A.1  In Coordinate Descent

In Section 2 we only discussed the expected progress that can be proven using the quadratic upper bound (1). Here we show how to derive the convergence rate by the standard arguments.

**Lemma A.1** (Proposed CD on strongly convex function—one step progress). *Let $f \colon \mathbb{R}^n \to \mathbb{R}$ $\mu$-strongly convex with coordinate-wise $L_i$-Lipschitz continuous gradient. Let $\boldsymbol{x}_k, \boldsymbol{x}_{k+1} \in \mathbb{R}^n$ denote two successive iterates generated by Algorithm 1, i.e. satisfying (2) and (4). Then*

$$\mathbb{E}\left[f(\boldsymbol{x}_{k+1}) - f^\star \mid \boldsymbol{x}_k\right] \leq (f(\boldsymbol{x}_k) - f^\star) \cdot (1 - \mu \alpha_k) \tag{14}$$

*where $f^\star = \min_{\boldsymbol{x} \in \mathbb{R}^n} f(\boldsymbol{x})$ and $\alpha_k = \alpha_k(\boldsymbol{p}_k)$ as in Lemma 2.1.*

*Proof.* By strong convexity

$$\frac{1}{2\mu} \|\nabla f(\boldsymbol{x}_k)\|_2^2 \geq f(\boldsymbol{x}_k) - f^\star, \tag{15}$$

and the claim follows directly from (4).  □

For example for $L_i$-based importance sampling, $\alpha_k \equiv \frac{1}{\mathrm{Tr}[\mathbf{L}]}$ (Example 2.2) and the statement simplifies to

$$\mathbb{E}\left[f(\boldsymbol{x}_{k+1}) - f^\star \mid \boldsymbol{x}_k\right] \leq (f(\boldsymbol{x}_k) - f^\star) \cdot \left(1 - \frac{\mu}{\mathrm{Tr}[\mathbf{L}]}\right) \tag{16}$$

in alignement with the results in [18, 31]. For the optimal sampling from Example 2.3 it holds $\alpha_k(\boldsymbol{p}_k^\star) = \frac{\|\nabla f(\boldsymbol{x}_k)\|_2^2}{\|\sqrt{\mathbf{L}}\nabla f(\boldsymbol{x}_k)\|_1^2}$. For instance for $\mathbf{L} = L \cdot \mathbf{I}_n$ equation (14) simplifies to

$$\mathbb{E}\left[f(\boldsymbol{x}_{k+1}) - f^\star \mid \boldsymbol{x}_k\right] \leq (f(\boldsymbol{x}_k) - f^\star) \cdot \left(1 - \frac{\mu \|\nabla f(\boldsymbol{x}_k)\|_2^2}{L \|\nabla f(\boldsymbol{x}_k)\|_1^2}\right). \tag{17}$$

By Cauchy-Schwarz $\|\nabla f(\boldsymbol{x}_k)\|_2^2 \leq \|\nabla f(\boldsymbol{x}_k)\|_1^2 \leq n \|\nabla f(\boldsymbol{x}_k)\|_2^2$, hence the expected one step progress (17) is always as least as good as for uniform sampling (16) (we assumed $\mathbf{L} = L \cdot \mathbf{I}_n$), but the optimal sampling could yield an $n$ times larger progress.

In Section 2 we argued that it is natural to always chose the best possible stepsize in (3), i.e. $\alpha_k = \alpha_k(\boldsymbol{p}_k)$. Interestingly, even with a fixed stepsize (the worst case $\alpha_k = \frac{1}{\mathrm{Tr}[\mathbf{L}]}$) the optimal sampling $\boldsymbol{p}_k^\star$ has a slight advantage over the fixed importance sampling $\boldsymbol{p}_\mathbf{L}$. (This effect is also demonstrated in the experiments, cf. Figure 2).

**Remark A.2.** *Let $\boldsymbol{p}_k^\star$ as in Example 2.3. Then for suboptimal $\alpha_k = \frac{1}{\mathrm{Tr}[\mathbf{L}]}$ it holds*

$$\mathbb{E}_{i_k \sim \boldsymbol{p}_k}\left[f(\boldsymbol{x}_{k+1}) \mid \boldsymbol{x}_k\right] \leq f(\boldsymbol{x}_k) - \frac{1}{2\,\mathrm{Tr}[\mathbf{L}]} \|\nabla f(\boldsymbol{x}_k)\|_2^2 \cdot \left(2 - \frac{\|\sqrt{\mathbf{L}}\nabla f(\boldsymbol{x}_k)\|_1^2}{\mathrm{Tr}[\mathbf{L}] \|\nabla f(\boldsymbol{x}_k)\|_2^2}\right). \tag{18}$$

The expression in the big bracket is bounded between 1 and $2 - \frac{1}{n}$. Hence the progress is always better then for the fixed distribution $\boldsymbol{p}_L$, but the speed-up is limited to a factor less than 2. In contrast, with the optimal $\alpha_k(\boldsymbol{p}_k^\star)$ the speed-up can reach a factor of $n$.

*Proof.* It suffices to just evaluate (3) with $\boldsymbol{p}_k^\star$ and $\alpha_k = \frac{1}{\mathrm{Tr}[\mathbf{L}]}$. $\qquad\square$

*Proof of Lemma 2.1.* For $c, d \geq 0$ consider $\min_\alpha -\alpha c + \frac{1}{2}\alpha^2 d$. This function is minimized for $\alpha^\star = \frac{c}{d}$ with value $-\frac{c^2}{2d} = -\frac{\alpha^\star c}{2}$. $\qquad\square$

*Proof of Example 2.2.* We evaluate (3) with $\boldsymbol{p}_\mathbf{L}$ and find

$$\mathbb{E}_{i_k \sim \boldsymbol{p}_k}\left[f(\boldsymbol{x}_{k+1}) \mid \boldsymbol{x}_k\right] \leq f(\boldsymbol{x}_k) - \alpha_k \|\nabla f(\boldsymbol{x}_k)\|_2^2 + \frac{1}{2}\alpha_k^2 \,\mathrm{Tr}\,[\mathbf{L}]\, \|\nabla f(\boldsymbol{x}_k)\|_2^2 \qquad (19)$$

which is minimized for $\alpha_k = \frac{1}{\mathrm{Tr}[\mathbf{L}]}$ as claimed. $\qquad\square$

*Proof of Example 2.3.* This is an immediate consequence of Lemma 2.4. The provided estimates follow from $\|\boldsymbol{y}\|_2^2 \leq \|\boldsymbol{y}\|_1^2 \leq \frac{1}{L_{\min}}\|\sqrt{\mathbf{L}}\boldsymbol{y}\|_2^2$ and $\|\sqrt{\mathbf{L}}\boldsymbol{y}\|_1^2 \leq \mathrm{Tr}\,[\mathbf{L}]\, \|\boldsymbol{y}\|_2^2$ by Cauchy-Schwarz, for $\boldsymbol{y} \in \mathbb{R}^n$. $\qquad\square$

*Proof of Lemma 2.4.* Without loss of generality, assume $\mathbf{L} = \mathbf{I}$. The claim is verified by checking the optimality conditions: $-[\boldsymbol{x}]_i^2 + \lambda[\boldsymbol{p}]_i^2 = 0$ for all $i \in [n]$ and Lagrange multiplier $\lambda \geq 0$. Thus $\lambda = \frac{[\boldsymbol{x}]_i^2}{[\boldsymbol{p}]_i^2}$ for all $i \in [n]$ and this is satisfied for the proposed solution $\frac{|\boldsymbol{x}|}{\|\boldsymbol{x}\|_1} \in \Delta^n$. $\qquad\square$

## A.2 In SGD

SGD methods are applicable to objective functions which decompose as a sum

$$f(\boldsymbol{x}) = \tfrac{1}{n}\sum_{i=1}^n f_i(\boldsymbol{x}). \qquad (20)$$

Previous work [22, 36, 37] has argued that the gradient based sampling $[\tilde{\boldsymbol{p}}_k^\star]_i = \frac{\|\nabla f_i(\boldsymbol{x}_k)\|_2}{\sum_{i=1}^n \|\nabla f_i(\boldsymbol{x}_k)\|_2}$ is also optimal in this setting. For the sake of completeness, we will now exhibit how this can be derived in the simplified setting where we assume $f$ to be $\mu$-strongly convex. The proof presented here is adapted from [17].

**Theorem A.3.** *Let $\mathcal{X} \in \mathbb{R}^d$ be a convex set, $f \colon \mathcal{X} \to \mathbb{R}$ $\mu$-strongly convex with the structure $f(\boldsymbol{x}) = \frac{1}{n}\sum_{i=1}^n f_i(\boldsymbol{x})$. Let $\{\boldsymbol{x}_k\}_{k \geq 0}$ denote a sequence of iterates satisfying*

$$\boldsymbol{x}_{k+1} := \Pi_\mathcal{X}\left(\boldsymbol{x}_k - \frac{\eta_k}{(n[\boldsymbol{p}_k]_{i_k})}\nabla f_{i_k}(\boldsymbol{x}_k)\right) \qquad (21)$$

*for stepsize $\eta_k = \frac{1}{\mu k}$, where index $i_k$ is chosen at random $i_k \sim \boldsymbol{p}_k$ for probability vector $\boldsymbol{p}_k \in \Delta^n$ and $\Pi_\mathcal{X}$ denotes the orthogonal projection onto $\mathcal{X}$.*

*(i) If $[\boldsymbol{p}_k]_i \equiv \frac{1}{n}$ for all $i \in [n]$ and $k$ (uniform sampling), then*

$$\mathbb{E}\left[f\left(\frac{1}{T}\sum_{k=0}^T \boldsymbol{x}_k\right) - f^\star\right] \leq \frac{B_2}{\mu^2 T}(1 + \log T). \qquad (22)$$

*(ii) If $[\boldsymbol{p}_k]_i = \frac{\|\nabla f_i(\boldsymbol{x}_k)\|_2}{\sum_{i=1}^n \|\nabla f_i(\boldsymbol{x}_k)\|_2} = [\tilde{\boldsymbol{p}}_k^\star]_i$, for $i \in [n]$ (optimal adaptive sampling), then*

$$\mathbb{E}\left[f\left(\frac{1}{T}\sum_{k=0}^T \boldsymbol{x}_k\right) - f^\star\right] \leq \frac{B_1^2}{\mu^2 T}(1 + \log T). \qquad (23)$$

*Where $B_1$ and $B_2$ are constants such that*

$$\frac{\sum_{i=1}^n \|\nabla f_i(\boldsymbol{x})\|_2}{n} \leq B_1 \qquad\qquad \frac{\sum_{i=1}^n \|\nabla f_i(\boldsymbol{x})\|_2^2}{n} \leq B_2 \qquad\qquad \forall \boldsymbol{x} \in \mathcal{X}. \qquad (24)$$

It is clear that $\frac{B_2}{n} \leq B_1^2 \leq B_2$ from Cauchy-Schwarz. Comparing the upper bound we see that the importance sampling based approach might be $n$-times faster in convergence.

*Proof.* As orthogonal projections contract distances we have

$$\|\boldsymbol{x}_{k+1} - \boldsymbol{x}^\star\|_2^2 \leq \|\boldsymbol{x}_k - \eta_k \frac{1}{n[\boldsymbol{p}_k]_{i_k}} \nabla f_{i_k}(\boldsymbol{x}_k) - \boldsymbol{x}^\star\|_2^2 \tag{25}$$

$$= \|\boldsymbol{x}_k - \boldsymbol{x}^\star\|_2^2 - \frac{2\eta_k}{n[\boldsymbol{p}_k]_{i_k}} \langle \boldsymbol{x}_k - \boldsymbol{x}^\star, \nabla f_{i_k}(\boldsymbol{x}_k)\rangle + \frac{\eta_k^2}{n^2[\boldsymbol{p}_k]_{i_k}^2} \|\nabla f_{i_k}(\boldsymbol{x}_k)\|_2^2. \tag{26}$$

Thus

$$\mathbb{E}\left[\|\boldsymbol{x}_{k+1} - \boldsymbol{x}^\star\|_2^2 \mid \boldsymbol{x}_k\right] \leq \|\boldsymbol{x}_k - \boldsymbol{x}^\star\|_2^2 - 2\eta_k \langle \boldsymbol{x}_k - \boldsymbol{x}^\star, \nabla f(\boldsymbol{x}_k)\rangle \tag{27}$$

$$+ \sum_{i=1}^{n} \frac{\eta_k^2}{n^2[\boldsymbol{p}_k]_{i_k}} \|\nabla f_{i_k}(\boldsymbol{x}_k)\|_2^2. \tag{28}$$

It can be observed that the right hand side is minimized for probabilities given as follows:

$$[\tilde{\boldsymbol{p}}_k^\star]_i := \frac{\|\nabla f_i(\boldsymbol{x}_k)\|_2}{\sum_{i=1}^{n} \|\nabla f_i(\boldsymbol{x}_k)\|_2}. \tag{29}$$

This justifies why these probabilities are denoted as optimal (cf. Section 3 and [22, 36, 37]).

Hence the expression becomes :

$$\mathbb{E}\left[\|\boldsymbol{x}_{k+1} - \boldsymbol{x}^\star\|_2^2 \mid \boldsymbol{x}_k\right] \leq \|\boldsymbol{x}_k - \boldsymbol{x}^\star\|_2^2 - 2\eta_k \langle \boldsymbol{x}_k - \boldsymbol{x}^\star, \nabla f(\boldsymbol{x}_k)\rangle$$

$$+ \eta_k^2 \left(\left(\frac{\sum_{i=1}^{n} \|\nabla f_i(\boldsymbol{x}_k)\|_2}{n}\right)^2 \right) \tag{30}$$

$$\leq \|\boldsymbol{x}_k - \boldsymbol{x}^\star\|_2^2 - 2\eta_k \left[f(\boldsymbol{x}_k) - f^\star + \frac{\mu}{2}\|\boldsymbol{x}_k - \boldsymbol{x}^\star\|_2^2\right]$$

$$+ \eta_k^2 \left(\frac{\sum_{i=1}^{n} \|\nabla f_i(\boldsymbol{x}_k)\|_2}{n}\right)^2 \tag{31}$$

where the last inequality follows from strong convexity. Now we rearrange the terms and utilize the choice of the step size $\eta_k := \frac{1}{\mu k}$:

$$2\eta_k \left[f(\boldsymbol{x}_k) - f^\star\right] \leq \eta_k^2 \left(\frac{\sum_{i=1}^{n} \|\nabla f_i(\boldsymbol{x}_k)\|_2}{n}\right)^2 + (1 - \mu\eta_k)\|\boldsymbol{x}_k - \boldsymbol{x}^\star\|_2^2$$

$$- \mathbb{E}\left[\|\boldsymbol{x}_{k+1} - \boldsymbol{x}^\star\|_2^2 \mid \boldsymbol{x}_k\right] \tag{32}$$

$$\left[f(\boldsymbol{x}_k) - f^\star\right] \leq \tfrac{1}{2}\eta_k \left(\frac{\sum_{i=1}^{n} \|\nabla f_i(\boldsymbol{x}_k)\|_2}{n}\right)^2 + \frac{1 - \mu\eta_k}{2\eta_k}\|\boldsymbol{x}_k - \boldsymbol{x}^\star\|_2^2$$

$$- \frac{1}{2\eta_k} \mathbb{E}\left[\|\boldsymbol{x}_{k+1} - \boldsymbol{x}^\star\|_2^2 \mid \boldsymbol{x}_k\right] \tag{33}$$

$$\left[f(\boldsymbol{x}_k) - f^\star\right] \leq \frac{1}{2\mu k} \left(\frac{\sum_{i=1}^{n} \|\nabla f_i(\boldsymbol{x}_k)\|_2}{n}\right)^2 + \frac{\mu(k-1)}{2}\|\boldsymbol{x}_k - \boldsymbol{x}^\star\|_2^2$$

$$- \frac{\mu k}{2} \mathbb{E}\left[\|\boldsymbol{x}_{k+1} - \boldsymbol{x}^\star\|_2^2 \mid \boldsymbol{x}_k\right] \tag{34}$$

If we compare the last equation and corresponding expression for uniform sampling then we see that the per iterate gain by the optimal sampling is approximately of the order of $n$ due to the term $\left(\frac{\sum_{i=1}^{n} \|\nabla f_i(\boldsymbol{x}_k)\|_2}{n}\right)^2$ in our case and $\frac{1}{n}\sum_{i=1}^{n} \|\nabla f_i(\boldsymbol{x}_k)\|_2^2$ in the uniform sampling.

We now take the expectation and sum the equation (34) for $k = 0, \dots T$ and we get the claim (this step is analogous as in [13]). $\qquad\square$

## B  Sampling

In this section we provide the remaining technical details regarding our proposed sampling scheme.

## B.1 On the solution of the optimization problem

In the proof of Theorem 3.2 we claimed that min and max in (7) can be interchanged. We will prove this now. This result will also be handy to describe the optimiality conditions of problem (7) in the proof of Theorem 3.4 below.

**Lemma B.1.** *It holds*

$$v_k = \min_{\boldsymbol{p} \in \Delta^n} \max_{\boldsymbol{c} \in C_k} \frac{V(\boldsymbol{p}, \boldsymbol{c})}{\|\boldsymbol{c}\|_2^2} \overset{(*)}{=} \max_{\boldsymbol{c} \in C_k} \min_{\boldsymbol{p} \in \Delta^n} \frac{V(\boldsymbol{p}, \boldsymbol{c})}{\|\boldsymbol{c}\|_2^2} = \max_{\boldsymbol{c} \in C_k} \frac{\|\sqrt{\mathbf{L}}\boldsymbol{c}\|_1^2}{\|\boldsymbol{c}\|_2^2}. \tag{35}$$

*Proof.* The third equality follows directly from Lemma 2.4. By transformation of the variable $[\boldsymbol{y}] := [\boldsymbol{c}]_i^2$ for $i \in [n]$ we can write the objective function as

$$\frac{V(\boldsymbol{p}, \boldsymbol{c})}{\|\boldsymbol{c}\|_2^2} = \frac{1}{\|\boldsymbol{y}\|_1} \cdot \sum_{i=1}^n \frac{L_i [\boldsymbol{y}]_i}{[\boldsymbol{p}]_i} =: \psi(\boldsymbol{p}, \boldsymbol{y}). \tag{36}$$

Let $Y \subset \mathbb{R}_{\geq 0}^n$ denote appropriately transformed set of constraints, $Y := C_k^2$. To prove $(*)$ we will now rely on Sion's minimax theorem [30, 12]. The function $\psi(\cdot, \boldsymbol{y})$ is convex in $\boldsymbol{p} \in \Delta^n$ and $\Delta^n$ is a compact convex subset of $\mathbb{R}^n$. Clearly, $Y$ is convex, and in order to apply the theorem it remains to show that $\psi(\boldsymbol{p}, \cdot)$ is quasi-concave. For establish this, it is enough to show that the level sets of $\psi(\boldsymbol{p}, \cdot)$ are convex. Let $\boldsymbol{u}, \mathbf{v} \in Y$ with $\psi(\boldsymbol{p}, \boldsymbol{u}) \geq \beta$, $\psi(\boldsymbol{p}, \mathbf{v}) \geq \beta$ for some $\beta \geq 0$. Then for any $\lambda \in [0, 1]$ it holds $\psi(\boldsymbol{p}, \lambda \boldsymbol{u} + (1 - \lambda)\mathbf{v}) \geq \beta$ as is verified as follows:

$$0 \leq \lambda \underbrace{\left[ \left( \sum_{i=1}^n \frac{[\boldsymbol{u}]_i L_i}{[\boldsymbol{p}]_i} \right) - \beta \|\boldsymbol{u}\|_1 \right]}_{\geq 0} + (1 - \lambda) \underbrace{\left[ \left( \sum_{i=1}^n \frac{[\mathbf{v}]_i L_i}{[\boldsymbol{p}]_i} \right) - \beta \|\mathbf{v}\|_1 \right]}_{\geq 0} \tag{37}$$

$$= \left( \sum_{i=1}^n \frac{\lambda [\boldsymbol{u}]_i L_i + (1 - \lambda)[\mathbf{v}]_i L_i}{[\boldsymbol{p}]_i} \right) - \beta \underbrace{\left( \lambda \|\boldsymbol{u}\|_1 + (1 - \lambda) \|\mathbf{v}\|_1 \right)}_{= \|\lambda \boldsymbol{u} + (1 - \lambda)\mathbf{v}\|_1}. \tag{38}$$

This proves the claim. $\square$

*Proof of Theorem 3.4 – Part I: Structure of the solution.* We will now proof that $\boldsymbol{c} \in C_k$ of the form

$$[\boldsymbol{c}]_i = \begin{cases} [\boldsymbol{u}_k]_i & \text{if } [\boldsymbol{u}_k]_i \leq \sqrt{L_i} m, \\ [\boldsymbol{\ell}_k]_i & \text{if } [\boldsymbol{\ell}_k]_i \geq \sqrt{L_i} m, \\ \sqrt{L_i} m & \text{otherwise,} \end{cases} \qquad \forall i \in [n], \tag{9}$$

where $m = \|\boldsymbol{c}\|_2^2 \cdot \|\sqrt{\mathbf{L}}\boldsymbol{c}\|_1^{-1}$ and probabilities $\boldsymbol{p} = \frac{\sqrt{\mathbf{L}}\boldsymbol{c}}{\|\sqrt{\mathbf{L}}\boldsymbol{c}\|_1}$ solve the optimization problem (7). By Lemma B.1 is suffices to consider

$$\arg\max_{\boldsymbol{c} \in C_k} \frac{\|\sqrt{\mathbf{L}}\boldsymbol{c}\|_1^2}{\|\boldsymbol{c}\|_2^2} = \arg\max_{\boldsymbol{c} \in C_k} \frac{\|\sqrt{\mathbf{L}}\boldsymbol{c}\|_1}{\|\boldsymbol{c}\|_2}. \tag{39}$$

We now write the Lagrangian of the problem on the right:

$$\mathcal{L}(\boldsymbol{c}, \boldsymbol{\lambda}, \boldsymbol{\mu}) = \frac{\|\sqrt{\mathbf{L}}\boldsymbol{c}\|_1}{\|\boldsymbol{c}\|_2} + \sum_{i=1}^n [\boldsymbol{\lambda}]_i ([\boldsymbol{u}_k]_i - [\boldsymbol{c}]_i) + \sum_{i=1}^n [\boldsymbol{\mu}]_i ([\boldsymbol{c}]_i - [\boldsymbol{\ell}_k]_i) \tag{40}$$

and derive the KKT conditions:

$$\frac{\partial \mathcal{L}}{\partial [\boldsymbol{c}]_i} = \frac{\sqrt{L_i} \|\boldsymbol{c}\|_2^2 - [\boldsymbol{c}]_i \|\sqrt{\mathbf{L}}\boldsymbol{c}\|_1}{\|\boldsymbol{c}\|_2^3} - [\boldsymbol{\lambda}]_i + [\boldsymbol{\mu}]_i \leq 0; \quad [\boldsymbol{c}]_i \geq 0; \quad [\boldsymbol{c}]_i \frac{\partial \mathcal{L}}{\partial [\boldsymbol{c}]_i} = 0; \tag{41}$$

$$\frac{\partial \mathcal{L}}{\partial [\boldsymbol{\lambda}]_i} = [\boldsymbol{u}_k]_i - [\boldsymbol{c}]_i \geq 0; \qquad\qquad\qquad [\boldsymbol{\lambda}]_i \geq 0; \quad [\boldsymbol{\lambda}]_i \frac{\partial \mathcal{L}}{\partial [\boldsymbol{\lambda}]_i} = 0; \tag{42}$$

$$\frac{\partial \mathcal{L}}{\partial [\boldsymbol{\mu}]_i} = [\boldsymbol{c}]_i - [\boldsymbol{\ell}_k]_i \geq 0; \qquad\qquad\qquad [\boldsymbol{\mu}]_i \geq 0; \quad [\boldsymbol{\mu}]_i \frac{\partial \mathcal{L}}{\partial [\boldsymbol{\mu}]_i} = 0; \tag{43}$$

For all non-binding constraints, the Lagrange multipliers are zero, and hence from the topmost equation see that it must hold $\sqrt{L_i}\,\|\boldsymbol{c}\|_2^2 - [\boldsymbol{c}]_i\|\sqrt{\mathbf{L}}\boldsymbol{c}\|_1 = 0$ (or equivalently $[\boldsymbol{c}]_i = \sqrt{L_i}m$) for all variables with non-binding constraints. Furthermore if $[\boldsymbol{c}]_i < \sqrt{L_i}m$), then $[\boldsymbol{\lambda}]_i$ must be positive, and hence the upper bound must be binding. And vice versa for the lower bounds. Clearly, the given $\boldsymbol{c}$ in (9) satisfies these conditions. By Lemma 2.4 we also have $\boldsymbol{p} = \boldsymbol{p}(\boldsymbol{c}) = \frac{\sqrt{\mathbf{L}}\boldsymbol{c}}{\|\sqrt{\mathbf{L}}\boldsymbol{c}\|_1}$ as claimed. □

## B.2   Algorithm

Here we argue on the correctness of Algorithm 4.

*Proof of Theorem 3.4 – Part II: Algorithm.*  We now show that Algorithm 4 indeed computes a solution of the form (9). For this, we have to show that performed optimization steps—the sorting in line 2 and the efficient comparisons in line 4 and 6—do not hamper the correctness for the algorithm. For clarity, we now introduce iteration indices for the quantities $\boldsymbol{c}_t$ (see main text), and $m_t$.

Suppose the check in line 4 is true, i.e. $[\boldsymbol{\ell}^{\mathrm{sort}}]_\ell > m_t$, where $m_t = \frac{\|\boldsymbol{c}_t\|_2^2}{\|\sqrt{\mathbf{L}}\boldsymbol{c}_t\|_1}$. Now we show $m_{t+1} \in [m_t, [\boldsymbol{\ell}^{\mathrm{sort}}]_\ell]$. The claim can easily be checked. Let $L_\tau$ denote the corresponding $L_i$-value, i.e. it holds $\sqrt{L_\tau}[\boldsymbol{\ell}^{\mathrm{sort}}]_\ell = [\boldsymbol{\ell}_k]_\tau$.

By assumption $[\boldsymbol{\ell}^{\mathrm{sort}}]_\ell > \frac{\|\boldsymbol{c}_t\|_2^2}{\|\sqrt{\mathbf{L}}\boldsymbol{c}_t\|_1}$, thus $[\boldsymbol{\ell}^{\mathrm{sort}}]_\ell \cdot \|\sqrt{\mathbf{L}}\boldsymbol{c}_t\|_1 + L_\tau[\boldsymbol{\ell}^{\mathrm{sort}}]_\ell^2 > \|\boldsymbol{c}_t\|_2^2 + L_\tau[\boldsymbol{\ell}^{\mathrm{sort}}]_\ell^2$ and consequently $m_{t+1} = \frac{\|\boldsymbol{c}_t\|_2^2 + L_\tau[\boldsymbol{\ell}^{\mathrm{sort}}]_\ell^2}{\|\sqrt{\mathbf{L}}\boldsymbol{c}_t\|_1 + L_\tau[\boldsymbol{\ell}^{\mathrm{sort}}]_\ell} < [\boldsymbol{\ell}^{\mathrm{sort}}]_\ell$. For to show $m_{t+1} > m_t$ we make use of the assumption $[\boldsymbol{\ell}^{\mathrm{sort}}]_\ell > \frac{\|\boldsymbol{c}_t\|_2^2}{\|\sqrt{\mathbf{L}}\boldsymbol{c}_t\|_1}$ in a similar way. Clearly, $L_\tau[\boldsymbol{\ell}^{\mathrm{sort}}]_\ell^2 \cdot \|\sqrt{\mathbf{L}}\boldsymbol{c}_t\|_1 > L_\tau[\boldsymbol{\ell}^{\mathrm{sort}}]_\ell \cdot \|\boldsymbol{c}_t\|_2^2$ and thus $\|\sqrt{\mathbf{L}}\boldsymbol{c}_t\|_1 \cdot \|\boldsymbol{c}_t\|_2^2 + L_\tau[\boldsymbol{\ell}^{\mathrm{sort}}]_\ell^2 \cdot \|\sqrt{\mathbf{L}}\boldsymbol{c}_t\|_1 > \|\sqrt{\mathbf{L}}\boldsymbol{c}_t\|_1 \cdot \|\boldsymbol{c}_t\|_2^2 + L_\tau[\boldsymbol{\ell}^{\mathrm{sort}}]_\ell \cdot \|\boldsymbol{c}_t\|_2^2$ which implies $m_{t+1} = \frac{\|\boldsymbol{c}_t\|_2^2 + L_\tau[\boldsymbol{\ell}^{\mathrm{sort}}]_\ell^2}{\|\sqrt{\mathbf{L}}\boldsymbol{c}_t\|_1 + L_\tau[\boldsymbol{\ell}^{\mathrm{sort}}]_\ell} > \frac{\|\boldsymbol{c}_t\|_2^2}{\|\sqrt{\mathbf{L}}\boldsymbol{c}_t\|_1} = m_t$.

The inequality $m_{t+1} \le [\boldsymbol{\ell}^{\mathrm{sort}}]_\ell$ implies that the chosen update does not interfere with any previously made decisions regarding lower bounds, as $m_{t+1} \le [\boldsymbol{\ell}^{\mathrm{sort}}]_i$ for $i = \ell+1, \ldots, n$ (with this notation, $n+1, \ldots, n$ just denotes the empty set). The opposite inequality $m_{t+1} \ge m_t$ implies that the chosen update does not interfere with any previously made decisions regarding upper bounds, as $m_{t+1} \ge [\boldsymbol{u}^{\mathrm{sort}}]_i$ for $i = 1, \ldots, u-1$.

If line 6 is executed and the check is true, i.e. $[\boldsymbol{u}^{\mathrm{sort}}]_u < m_t$, then it can be shown that $m_{t+1} \in [[\boldsymbol{u}^{\mathrm{sort}}]_u, m_t]$ by analogous arguments. □

## B.3   Competitive Ratio

*Proof of Lemma 3.5.*  The proof of this lemma is immediate from the definition:

$$\rho_k = \max_{\boldsymbol{c}\in C_k} \frac{V(\hat{\boldsymbol{p}}, \boldsymbol{c})}{\|\boldsymbol{c}\|_2^2} \cdot \frac{\|\boldsymbol{c}\|_2^2}{\|\sqrt{\mathbf{L}}\boldsymbol{c}\|_1^2} \le \max_{\boldsymbol{c}\in C_k} \frac{V(\hat{\boldsymbol{p}}, \boldsymbol{c})}{\|\boldsymbol{c}\|_2^2} \cdot \max_{\boldsymbol{c}\in C_k} \frac{\|\boldsymbol{c}\|_2^2}{\|\sqrt{\mathbf{L}}\boldsymbol{c}\|_1^2} \le \frac{v_k}{w_k} \,. \tag{44}$$

where $w_k := \min_{\boldsymbol{c}\in C_k} \frac{\|\sqrt{\mathbf{L}}\boldsymbol{c}\|_1^2}{\|\boldsymbol{c}\|_2^2}$. The claimed upper bound $w_k \le v_k$ follows by the observation $v_k \overset{(35)}{=} \max_{\boldsymbol{c}\in C_k} \frac{\|\sqrt{\mathbf{L}}\boldsymbol{c}\|_1^2}{\|\boldsymbol{c}\|_2^2}$. □

*Proof of Lemma 3.6.*  As we have relative accuracy, it holds $[C_k]_i \cap \gamma[C_k]_i = \emptyset$ and $\gamma^{-1}[C_k]_i \cap [C_k]_i = \emptyset$, for all $i \in [n]$. Let $\boldsymbol{c}^\star \in C_k$ denote the vector for which the maximum is attained and let $\hat{\boldsymbol{c}} \in C_k$ be such that $\hat{\boldsymbol{p}} = \frac{\sqrt{\mathbf{L}}\hat{\boldsymbol{c}}}{\|\sqrt{\mathbf{L}}\hat{\boldsymbol{c}}\|_1}$. It holds $V(\hat{\boldsymbol{p}}, \boldsymbol{c}^\star) \le V(\hat{\boldsymbol{p}}, \boldsymbol{c})$ for all $\boldsymbol{c} \in \gamma C_k$ by monotonicity in each coordinate, especially $V(\hat{\boldsymbol{p}}, \boldsymbol{c}^\star) \le V(\hat{\boldsymbol{p}}, \gamma\hat{\boldsymbol{c}})$. And similarly $\|\sqrt{\mathbf{L}}\boldsymbol{c}^\star\|_1^2 \ge \|\gamma^{-1}\sqrt{\mathbf{L}}\hat{\boldsymbol{c}}\|_1^2$. Thus

$$\rho_k \le \frac{V(\hat{\boldsymbol{p}}, \gamma\hat{\boldsymbol{c}})}{\|\gamma^{-1}\sqrt{\mathbf{L}}\hat{\boldsymbol{c}}\|_1^2} = \frac{\gamma^2 V(\hat{\boldsymbol{p}}, \hat{\boldsymbol{c}})}{\gamma^{-2}\|\sqrt{\mathbf{L}}\hat{\boldsymbol{c}}\|_1^2} = \frac{\gamma^2}{\gamma^{-2}} \,. \tag{45}$$

which proves the claim. □

# C  Safe Gradient Bounds in the Proximal Setting

*Proof of Lemma 4.1.*  Observe

$$
\begin{aligned}
\nabla f_i(\boldsymbol{x}_{k+1}) - \nabla f_i(\boldsymbol{x}_k) &= \nabla_x h_i(\boldsymbol{a}_i^\top \boldsymbol{x}_{k+1}) - \nabla_x h_i(\boldsymbol{a}_i^\top \boldsymbol{x}_k) \\
&= \boldsymbol{a}_i\big(\nabla h_i(\boldsymbol{a}_i^\top \boldsymbol{x}_{k+1}) - \nabla h_i(\boldsymbol{a}_i^\top \boldsymbol{x}_k)\big) \\
&= \boldsymbol{a}_i\big(\boldsymbol{a}_i^\top \boldsymbol{x}_{k+1} - \boldsymbol{a}_i^\top \boldsymbol{x}_k\big)\nabla^2 h_i(\boldsymbol{a}_i^\top \tilde{\boldsymbol{x}}) \\
&= \boldsymbol{a}_i\big(\boldsymbol{a}_i^\top (\boldsymbol{x}_{k+1} - \boldsymbol{x}_k)\nabla^2 h_i(\boldsymbol{a}_i^\top \tilde{\boldsymbol{x}})\big) \\
&= \boldsymbol{a}_i\big(\boldsymbol{a}_i^\top \gamma_k \boldsymbol{a}_{i_k}\nabla^2 h_i(\boldsymbol{a}_i^\top \tilde{\boldsymbol{x}})\big) \\
&= \gamma_k \nabla^2 h_i(\boldsymbol{a}_i^\top \tilde{\boldsymbol{x}})\langle \boldsymbol{a}_i, \boldsymbol{a}_{i_k}\rangle \boldsymbol{a}_i \quad \forall\, i \neq i_k\,,
\end{aligned}
\tag{46}
$$

Equation (46) comes from the mean value theorem which says for continuous function $f$ in closed intervals $[a,b]$ and differentiable on open intervals $(a,b)$, there exists a point $c$ in $(a,b)$ such that :

$$
f'(c) = \frac{f(b) - f(a)}{b - a}\,.
\tag{47}
$$

$\square$

In Section 4 we have discussed practical safe upper and lower bounds $\boldsymbol{u}, \boldsymbol{\ell}$ that can be maintained efficiently during optimization, also for the SGD setting (finite sum objective). We now argue that such bounds can also be extended to proximal SGD settings.

We see from Lemma 4.1 that tracking the norm of the gradient of each function can be done easily for simple updates as given in Lemma 4.1. The approximate update of the component wise gradient norms for more composite problems can also be done by a little modification, but it is definitely not as trivial as in the case of coordinate descent. For example, consider a proximal type of update as $\boldsymbol{x}_{k+1} = prox_{\eta_k g}\big(\boldsymbol{x}_k - \eta_k \cdot \boldsymbol{a}_{i_k}\nabla f_{i_k}(\boldsymbol{a}_{i_k}^\top \boldsymbol{x}_k)\big)$ which implies that $\boldsymbol{x}_{k+1} \in \boldsymbol{x}_k - \eta_k \cdot \boldsymbol{a}_{i_k}\nabla f_{i_k}(\boldsymbol{a}_{i_k}^\top \boldsymbol{x}_k) - \eta_k \partial g(\boldsymbol{x}_{k+1})$ and thus $\boldsymbol{x}_{k+1} \in \boldsymbol{x}_k + \gamma_k \cdot \boldsymbol{a}_{i_k} - \eta_k \partial g(\boldsymbol{x}_{k+1})$. If we denote the progress made in the $k$-th iteration of the algorithm as $\delta_k$ then the progress equals $\delta_k = \gamma_k\, \boldsymbol{a}_{i_k} - \eta_k \boldsymbol{\alpha}_k$ where $\boldsymbol{\alpha}_k \in \partial g(\boldsymbol{x}_{k+1})$. To approximate the gradient we will need to compute two dot products. The first one is $\langle \boldsymbol{a}_i, \boldsymbol{a}_{i_k}\rangle$ and the second one is $\langle \boldsymbol{a}_{i_k}, \boldsymbol{\alpha}_k\rangle$. Since $\boldsymbol{\alpha}_k$ is usually small, hence even approximating $\langle \boldsymbol{a}_{i_k}, \boldsymbol{\alpha}_k\rangle$ with $\|\boldsymbol{a}_{i_k}\|\|\boldsymbol{\alpha}_k\|$ doesn't affect the upper and bounds too much and the main contribution in error comes from the approximation of the scalar product $\langle \boldsymbol{a}_i, \boldsymbol{a}_{i_k}\rangle$.