[Reviews · NeurIPS 2017]

Reviewer 1



The authors present a "safe" adaptive importance sampling strategy for coordinate descent and stochastic gradient methods. Based on lower and upper bounds on the gradient values, an efficient approximation of gradient based sampling is proposed. The method is proven to be the best strategy with respect to the bounds, always better than uniform or fixed importance sampling and can be computed efficiently for negligible extra cost. Although adaptive importance sampling strategies have been previously proposed, the authors present a novel formulation of selecting the optimal sampling distribution as a convex optimization problem and present an efficient algorithm to solve it. This paper is well written and a nice contribution to the study of importance sampling techniques. Comments: Proof of Lemma 2.1 -- seems to be missing a factor of 2 in alpha^*. Example 3.1 - In (7) you want to maximize, so in example 3.1, it would appear that setting c to the upper or lower bound is better than using uniform sampling. Is that what this example is intending to show? It is confusing with the statement directly prior claiming the naive approach of setting c to the upper or lower bound can be suboptimal. Line 4 of Algorithm 4 - given that m = max(l^{sort}), will this condition ever be satisfied? Line 7 of Algorithm 4 - should be u^{sort} instead of c^{sort}? I think the numerical results could benefit from comparisons to other adaptive sampling schemes out there (e.g., [2], [5],[21]), and against fixed importance sampling with say a non-uniform distribution. Why are there no timing results for SGD? Title of the paper in reference [14] is incorrect. Add reference: Csiba and Richtarik. "Importance Sampling for Minibatches" 2016 on arXiv. ============ POST REBUTTAL ============ I have read the author rebuttal and I thank the authors for addressing my questions/comments. I think this paper is a clear accept.

Reviewer 2



I think this paper is interesting and provide some good insight for RCD and SGD algorithms. Conclusion 2 in Theorem 3.2 is very useful because it suggests that the new sampling method is better than L based sampling which is a commonly used strategy. I have the following comments which I want the authors to consider. 1. In Line 85, in the denominator of p^*_i, it should be x_k not x. 2. In Line 135, you may want to remove the subscript k for \hat p and \hat c just to be consistent. 3. I cannot see any tick, label, number and word in all figures, which are too small. Please move some of them to appendix to gain space so that you can show larger figures. 4. If possible, please provide a result similar to Conclusion 2 in Theorem 3.2 but comparing with uniform sampling (p=1/n) instead of L_i based method 5. Please remark on how will this new sampling method affect the total complexity of RCD or SGD. For example, what will be the total complexity (including the O(n) time for updating l_k and u_k) for RCD to find an epislon-optimal solution?

Reviewer 3



This paper focuses on safe adaptive importance sampling, which is a popular topic in the community at the moment. Some of the key features of this work are that the proposed method is generic and can be integrated with many existing algorithms, and there are both theoretical results and practical experiments to support the proposed strategy. One issue with the paper that should be addressed is the figures in Section 5, which are very difficult to see properly. These figures should be made larger, and the labels also made larger, so that the reader can see them properly.